# The role of bacteria in wastewater treatment and the impact of treated wastewater on riverine bacterial ecosystems

**Akifumi Nishida**[1]***, Mayuko Nakagawa**[2]**, Masayuki Yamamura**[3]

**1** Department of Molecular Microbiology, Tokyo University of Agriculture, Tokyo, Japan, **2** Department of Earth and Planetary Sciences, Institute of Science Tokyo, Tokyo, Japan, **3** Department of Computer Science, Institute of Science Tokyo, Kanagawa, Japan

* an207919@nodai.ac.jp

## Abstract

Spatiotemporal analysis of bacterial communities in wastewater can explain the role of bacteria in removing organic matter, phosphorus, and nitrogen as well as the impact of wastewater treatment on riverine ecosystems. This study investigated the bacterial dynamics within an anaerobic–anoxic–oxic ($A_2O$) wastewater treatment plant (WWTP), comprising anaerobic, anoxic, and aerobic tanks, in Tokyo and its impact on the receiving Tama River. 16S rRNA gene analysis and ion chromatography were used to monitor bacterial composition and nutrient concentrations, respectively, to assess the effectiveness of nutrient removal across seasonal temperature variations, and the influence of treated wastewater on riverine bacterial communities. The $A_2O$ process effectively removed nutrients, but the nitrification efficiency was affected by temperature, with decreased temperatures correlating with reduced *Nitrospira* abundance, while nitrate concentrations increased due to higher influent ammonium loads. The WWTP bacterial community exhibited a polarized structure with a dominant core community that was stable over time and across tanks. The abundance of bacterial DNA introduced into the river via treated wastewater decreased downstream, indicating the spatial attenuation of the wastewater treatment impact on the riverine ecosystem. This study demonstrated the temperature sensitivity of WWTP processes, and the transient impact of treated wastewater discharge on river bacterial communities, thereby emphasizing the importance of understanding these dynamics for effective environmental conservation.

## Introduction

Wastewater treatment plants (WWTPs) utilize bacteria to remove nutrients from wastewater and help maintain the ecosystems of rivers and oceans where the treated wastewater is discharged. Phosphorus and nitrogen are essential for the growth of life, and when wastewater, containing these nutrients, is released into the

**Data availability statement:** Sequencing data were deposited in the National Center for Biotechnology Information (Accession: PRJNA1235227).

**Funding:** This study was financially supported by the Nissei Foundation (http://www.nihonseimei-zaidan.or.jp/kankyo/02.html) in the form of a grant (2018-17) received by AN. This study was also financially supported by JSPS KAKENHI (https://www.jsps.go.jp/english/e-grants) in the form of grants (22K17999 and 23KK0192) received by AN. This study was also financially supported by the Tokyu Foundation (https://foundation.tokyu.co.jp) in the form of a grant (2018-16) received by MY. The funders had no role in study design, data collection and analysis, decision to publish, or preparation of the manuscript.

**Competing interests:** The authors have declared that no competing interests exist.

environment without treatment, it causes environmental stress [1–4]. Water quality deteriorates at the treated wastewater discharge point, causing the water temperature and nutrient concentrations to increase and dissolved oxygen to decrease. Many ecological studies have reported that this change has a negative effect on aquatic invertebrates and fish [5–9].

For example, the anaerobic–anoxic–oxic ($A_2O$) process, which is common in WWTPs, can remove organic matter, nitrogen, and phosphorus from wastewater. The $A_2O$ process comprises three consecutive tanks—anaerobic, anoxic, and aerobic (oxic). Phosphorus-accumulating organisms (PAOs) release phosphorus in the anaerobic environment, absorb more phosphorus than they release in the aerobic environment, and finally, phosphorus is removed as sludge containing the PAOs. In the aerobic tank, nitrification occurs by autotrophic bacteria, and ammonia ($NH_3$) is first oxidized to nitrite ($NO_2^-$) by ammonia-oxidizing bacteria, and then nitrite ($NO_2^-$) is oxidized to nitrate ($NO_3^-$) by nitrite-oxidizing bacteria. Internal recycling occurs from the aerobic to the anoxic tank, and nitrite and/or nitrate are used as electron acceptors. Finally, nitrate is reduced to nitrogen gas, although conditions such as temperature, dissolved oxygen, pH, and supplied ammonium concentration can lead to the generation of the greenhouse gas nitrous oxide [10–12]. It is important to analyze how the chemical composition of wastewater is altered by bacteria during the $A_2O$ process so as to control the nutrients in the wastewater.

Spatiotemporal measurements of bacteria and chemical compositions in wastewater treatment processes provide insights into how the process controls bacteria-mediated nutrient conditions. Furthermore, it is important to analyze the bacterial communities, both in the wastewater treatment process and the receiving river for understanding the impact of the discharged treated-wastewater on the river ecosystem. While most studies on the effects of treated wastewater on rivers have focused on aquatic invertebrates and fish, only some have examined the effects on bacteria that play a fundamental role in nutrient cycling. At WWTPs in Chicago, USA, an increase in inorganic nutrients and a decrease in bacterial community diversity were observed when treated wastewater was discharged into rivers [13,14]. Antimicrobial resistance genes have also been detected in receiving rivers, raising concerns about the effect of the resistome (set of all antibiotic resistance genes) on bacterial communities [15]. In the Tama River in Japan, the discharge of treated wastewater caused major changes in the chemical composition of the river and deterministic changes in the structure of the bacterial community [16,17]. Given the significant impact of treated wastewater on riverine ecosystems demonstrated in these studies, clarifying the spatial dynamics of bacteria introduced via wastewater along the river is crucial for assessing potential ecological risks, such as the dispersal of antibiotic-resistant bacteria.

In this study, we hypothesized that seasonal temperature variations act as a deterministic factor reducing the abundance of key nitrifiers, such as *Nitrospira*, within the $A_2O$ process, thereby altering the bacterial composition discharged into the environment. Furthermore, we hypothesized that while treated wastewater discharge introduces distinct bacterial taxa into the receiving river, their persistence would be

spatially limited due to physical dispersion and environmental removal processes. To test these hypotheses, we monitored the bacterial communities in the anaerobic, anoxic, and aerobic tanks of an $A_2O$ process in Tokyo and the receiving Tama River from summer through winter using 16S rRNA gene analysis alongside chemical composition analysis using ion chromatography.

## Materials and methods

### Ethics statement

This study was conducted with the formal approval of the Bureau of Sewerage, Tokyo Metropolitan Government. Access to the wastewater treatment plant and the use of samples were approved by the Director of the Environmental Management Section, Facilities Management Division, on August 1, 2018. The publication of the research results was officially authorized by the Director of the Facilities Management Section, Technology Department, Regional Sewerage Office, on March 17, 2025. As this study analyzed bacterial communities in wastewater and river water and did not involve human participants or vertebrate animals, specific informed consent from individuals was not required.

### Sample site and sample collection

The urban WWTP in Tokyo, using the $A_2O$ treatment process, discharges treated water into the Tama River (Fig 1). The $A_2O$ treatment process consisted of an anaerobic tank, an anoxic tank, and an aerobic tank; the capacity and residence time of each tank are shown in S1 Table. Operational parameters of the $A_2O$ process were provided by the Tokyo Metropolitan Government Bureau of Sewerage. These parameters were the oxidation-reduction potential (ORP) in the anaerobic and anoxic tanks, mixed liquor suspended solids (MLSS), dissolved oxygen (DO), and influent temperature (monitored monthly) (S2 Table). Samples of wastewater (300 mL) were collected in sterile polypropylene bottles (AS ONE, Osaka, Japan) from the outlet at the end of each tank once every two weeks from August 2018 to February 2019. The samples were stored on ice and DNA was isolated within 3 h. River water samples were collected from the Tama River at site S3 (treated wastewater released), site S1 (1 km upstream of site S3), site S4 (2 km downstream of site S3), and site S2 (treated wastewater). The specific geographic coordinates of all sampling sites are provided in S3 Table. The sampling amount and storage method were the same as when the wastewater was collected.

### DNA isolation, polymerase chain reaction (PCR) amplification, sequencing, and data availability

Genomic DNA was isolated from the wastewater and river water samples using a DNeasy PowerWater Sterivex DNA Isolation Kit (Qiagen, Hilden, Germany). PCR amplification of the 16S rRNA gene V3 and V4 variable regions and MiSeq sequencing (Illumina, San Diego, CA, USA) using the MiSeq Reagent Kit v3 (600 cycles) (Illumina, San Diego, CA, USA) were performed as previously described [18]. Sequencing data were deposited in the National Center for Biotechnology Information (Accession: PRJNA1235227).

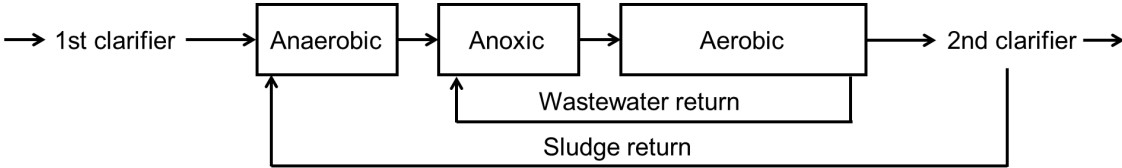

**Fig 1. Anaerobic-Anoxic-Aerobic ($A_2O$) treatment process.** A portion of wastewater is returned from the aerobic tank to the anoxic tank, and the sludge from the 2nd clarifier is returned to the anaerobic tank.

## Microbiome analysis based on 16S rRNA gene sequences

The basic method used for microbiome analysis has been described previously [17]. The QIIME2 Docker software (qiime2–2019.10) was used to analyze the 16S rRNA gene sequences [19]. A total of 300 nucleotide (nt) paired-end reads were trimmed using the *qiime dada2 denoise-paired* command (31–299 nt forward and 17–278 nt reverse). The reads were taxonomically classified with 99% amplicon sequence variant (ASV) data using SILVA 132 [20] and the *qiime feature-classifier extract-reads* command (CTACGGGGGGCAGCAG for forward and GGACTACCGGGGTATCT for reverse). α-diversity was calculated using QIIME2 with a sampling depth of 10,000 reads. The weighted UniFrac distance was calculated using QIIME2 for Principal coordinate analysis (PCoA). PCoA was performed with environmental variables using the *envfit* function of the R vegan package (version 2.6–10) [21,22], and environmental variable vectors significantly correlated with ordination axes ($p < 0.05$) were displayed.

To evaluate the natural spatial variation of bacterial communities in the Tama River, we utilized 16S rRNA gene sequencing data from two upstream sites (upstream 1 and upstream 2) collected on August 31, 2018, as described in our previous study [17]. These sites are located upstream of the study area and are separated by approximately 8.5 km. The similarity of the bacterial communities between these two reference sites was analyzed to establish a baseline for natural riverine heterogeneity.

## Statistical analysis

The normality of the data distribution was assessed using the Shapiro-Wilk test. Since the bacterial relative abundance data did not follow a normal distribution, non-parametric tests were employed.

To compare bacterial communities across river sites (S1–S4) while accounting for temporal variability (sampling dates), the Friedman test (a non-parametric alternative to repeated measures ANOVA) was performed using the friedmanchisquare function from the SciPy library (version 1.14.1) in Python [23]. Post-hoc pairwise comparisons were conducted using the Nemenyi test with the *posthoc_nemenyi_friedman* function from the scikit-posthocs library (version 0.11.3) [24].

For beta-diversity analysis, the Weighted UniFrac distance was selected as the similarity metric because it accounts for both the phylogenetic relatedness of bacterial sequences and their relative abundances. Differences in community structures between tanks were tested using Permutational Multivariate Analysis of Variance (PERMANOVA) based on Weighted UniFrac distances. PERMANOVA was conducted using the *adonis* function in the R vegan package (version 2.6–10) with 999 permutations.

To assess relationships between ion concentrations and temperature, Spearman's rank correlation analysis was performed using the *spearmanr* function from the SciPy library (version 1.6.3) in Python. For all statistical tests, a p-value of < 0.05 was considered statistically significant.

## Ion chromatography analysis in A$_2$O treatment system

Water samples for ion chromatography analysis were collected using 50 mL syringes and filtered with membrane syringe filters of 0.20 μm pore size (DISMIC–25AS; Advantec Toyo Kaisha, Tokyo, Japan) as previously described [17]. Anion concentrations were measured by ion chromatography using a Shimadzu Ion Chromatograph (Shimadzu, Kyoto, Japan) equipped with a Shodex SI-90 4E anion column (Showa Denko, Tokyo, Japan). A Shim-pack IC-C4 (Shimadzu, Kyoto, Japan) without a cation suppressor was used.

## Results and discussion

### Bacterial and chemical composition in A$_2$O treatment process

Analysis of the spatial chemical composition of wastewater from the anaerobic, anoxic, and aerobic tanks in the A$_2$O treatment process revealed significant alterations in nitrogen compounds (Fig 2A, S1 Dataset). Specifically, the process

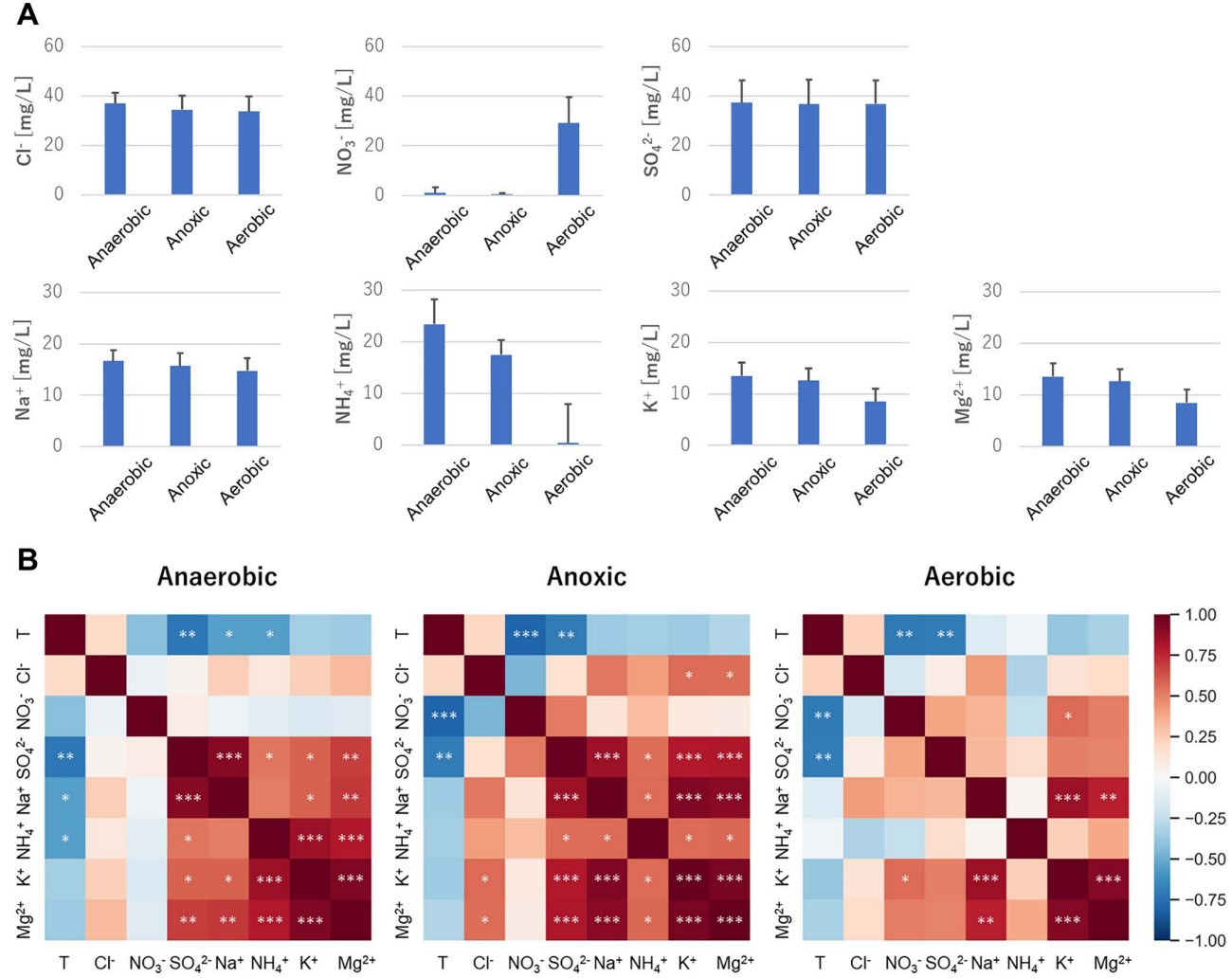

**Fig 2. Chemical composition and correlations in A₂O treatment process. (A)** Ion concentrations in anaerobic, anoxic, and aerobic tanks. **(B)** Correlations between ion concentrations and temperature **(T)**. * indicates $p < 0.05$, ** indicates $p < 0.01$, and *** indicates $p < 0.001$.

achieved a high ammonium removal efficiency of 98.4±5.8% (mean±SD). This effective nitrification was characterized by a sharp decrease in ammonium ($NH_4^+$) and a concurrent increase in nitrate ($NO_3^-$) within the aerobic tank. Bacterial metabolic activities contribute to these chemical changes, and their composition showed that major phyla were Proteobacteria (mean relative abundance 0.359±0.059 SD), Bacteroidetes (0.203±0.030 SD), Chloroflexi (0.109±0.050 SD), Actinobacteria (0.069±0.014 SD), Acidobacteria (0.061±0.023 SD), Verrucomicrobia (0.061±0.021 SD), Patescibacteria (0.031±0.014 SD), Firmicutes (0.028±0.037 SD), Epsilonbacteraeota (0.027±0.032 SD), and BRC1 (0.014±0.006 SD) (Fig 3A, S2 Dataset). At the phylum level, members of the microbiota were the same in different tanks and at different times. The alpha diversity of the bacterial community did not show any significant differences between the tanks (Fig 3B) (Student's t-test, $p > 0.05$). In contrast, PCoA based on the weighted UniFrac distance revealed that the bacterial community structure was strongly influenced by temporal variations, as samples from the same period clustered closely (Fig 3C). This seasonal shift was statistically supported by the environmental vector analysis (*envfit*), which identified temperature as a significant driver ($p < 0.05$) correlated with the ordination axes. However, even within this dominant temporal

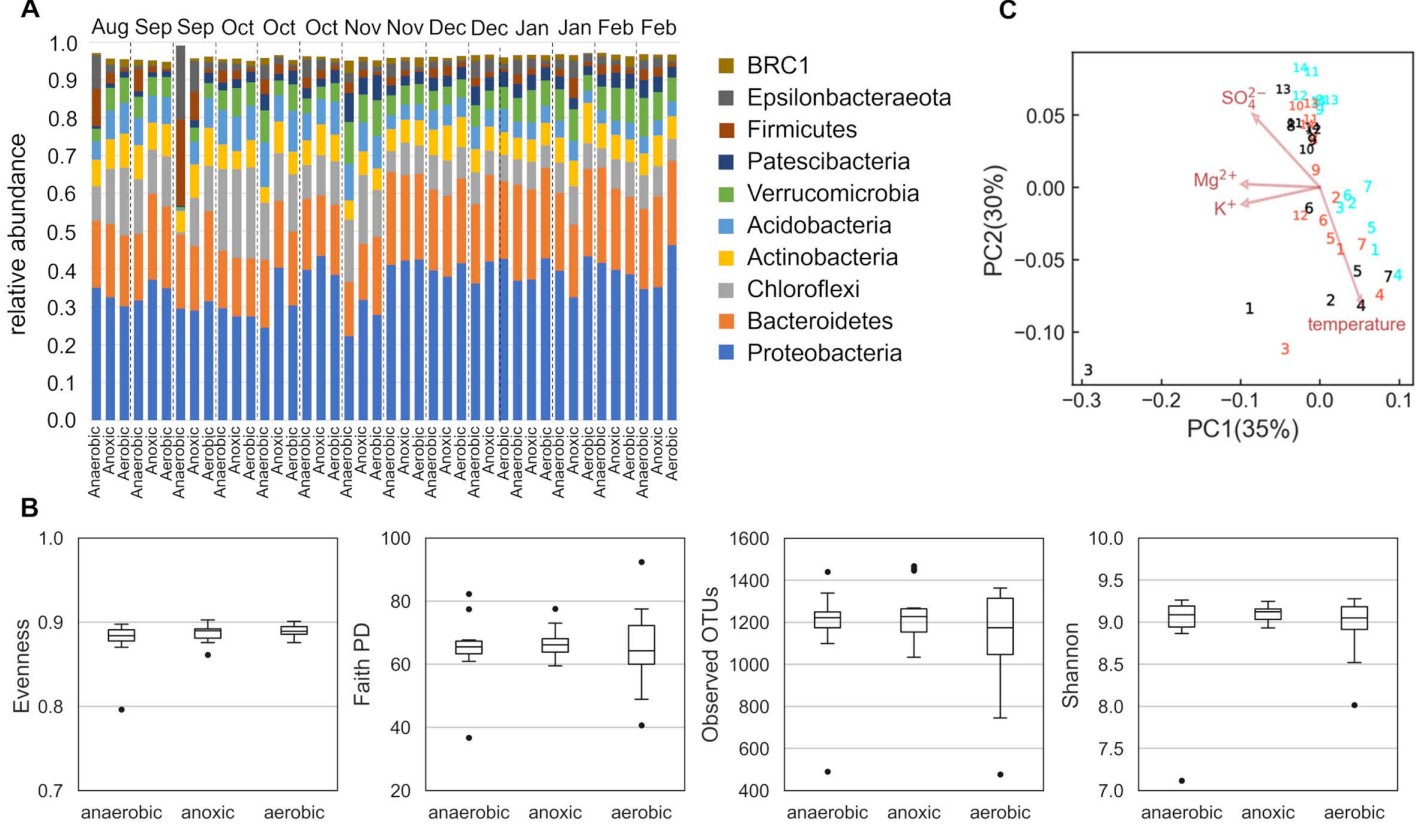

**Fig 3. Bacterial composition in A$_2$O treatment process.** (A) bacterial composition at phylum level with top 10 bacteria. **(B)** Comparison of α-diversity indices (Evenness, Faith's PD, Observed OTUs, Shannon) between tanks. **(C)** PCoA based on weighted UniFrac distance of microbiota. The numbers on the markers indicate the sampling period, with 1 representing summer and the numbers progressing towards 14 representing winter. The colors of the markers indicate the sample source: black for the anaerobic tank, orange for the anoxic tank, and cyan for the aerobic tank.

framework, the bacterial communities in the anaerobic/anoxic and aerobic tanks were different (anaerobic vs. aerobic: $p < 0.005$, F = 3.46, $R^2 = 0.12$; anoxic vs. aerobic: $p < 0.05$, F = 2.45, $R^2 = 0.12$, PERMANOVA).

Regarding the correlations between the chemical ion concentrations in the wastewater treatment process, there were positive correlations between sulfate ($SO_4^{2-}$), sodium ($Na^+$), ammonium ($NH_4^+$), potassium ($K^+$), and magnesium ($Mg^{2+}$) in the anaerobic and anoxic tanks ([Fig 2B]). $Na^+$, $K^+$, and $Mg^{2+}$ were also positively correlated in the aerobic tank. Temperature was negatively correlated with $SO_4^{2-}$ in all tanks and negatively correlated with $NO_3^-$ in the aerobic tank. As the temperature decreased, the inflow of $NH_4^+$ into the wastewater treatment plant increased; however, the activity of nitrifying bacteria may have decreased as $NO_3^-$ production also decreased. Other WWTPs reported a similar decrease in nitrification performance during the cold winter season and $NH_4^+$ amount increased [25–27].

The functional guilds of the WWTP, categorized according to Dueholm et al. [28], are shown in [Fig 4]. A key observation was the decline in the relative abundance of *Nitrospira*, a primary nitrite-oxidizing bacterium, during colder months, despite the increase in $NO_3^-$ concentrations in the aerobic tank. This apparent discrepancy implies that the nitrification process was limited by substrate availability rather than bacterial abundance. As indicated by the high ammonium removal efficiency (98.4 ± 5.8%), the nitrification process was substrate limited. This means the nitrifying community, even with reduced relative abundance in winter, retained sufficient capacity to completely oxidize the influent ammonium. Therefore,

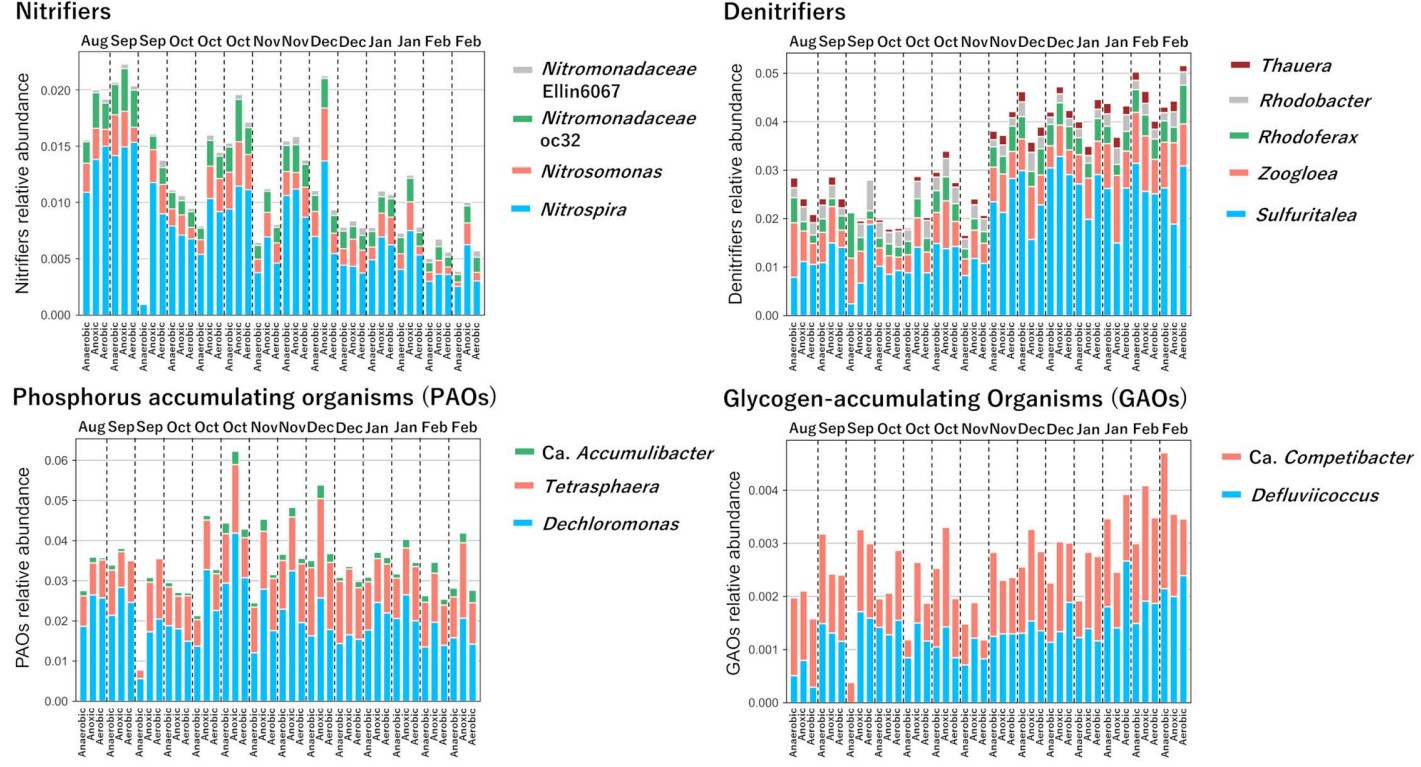

**Fig 4. The functional guilds in A$_2$O treatment process.**

the observed increase in NO$_3^-$ concentration was not due to increased bacterial activity per se but was a stoichiometric result of the increased influent NH$_4^+$ load during winter being fully converted to nitrate.

To assess whether factors other than temperature influenced the community and nitrification, we analyzed ancillary variables including ORP, MLSS, and DO (S2 Table). Throughout the study period, DO concentrations in the aerobic tank were generally maintained above 1.8 mg/L, indicating that oxygen availability was not a limiting factor for nitrification. Interestingly, MLSS concentrations increased from summer (1,340 mg/L in August) to winter (1,910 mg/L in February). This suggests an operational strategy to compensate for reduced bacterial activity in colder temperatures by increasing the total biomass. However, despite this increase in biomass, the relative abundance of *Nitrospira* decreased, and nitrate production dynamics shifted. This reinforces the conclusion that temperature-dependent reaction kinetics, rather than biomass quantity or oxygen limitation, were the primary drivers of the observed seasonal changes.

Conversely, distinct dynamics were observed in the denitrifying community. *Sulfuritalea*, a genus capable of denitrification, increased in relative abundance as temperatures dropped (Fig 4). Consistent with its reported prevalence in cold environments [29,30], this suggests a compensatory ecological response where *Sulfuritalea* proliferated to fill the niche of cold-sensitive heterotrophic denitrifiers. However, the net accumulation of NO$_3^-$ indicates that despite this compensatory shift, the overall denitrification capacity was insufficient to remove the increased nitrate generated from the high winter ammonium load. Thus, the winter condition was characterized by complete nitrification driven by high influent load, coupled with kinetically limited denitrification.

## Spatiotemporal analysis of bacterial communities in A$_2$O treatment process

To determine whether bacteria were transient or frequently detected (core) in the A$_2$O treatment process, ASVs commonly detected across time points were analyzed (Fig 5A). The number of ASVs detected only once was the largest, but many ASVs were detected in common throughout the entire period, suggesting that the bacteria in the wastewater treatment plant were polarized between the transient and core. As regards the number of ASVs, frequently observed ASVs (observed in >12 of 14 samples) were the dominant community, making up >70% of the total (Fig 5B). This polarization between transient and core ASVs and the dominance of core ASVs have the same characteristics as the microbiota of the Danish wastewater treatment plant [31]. However, the bacterial ecosystem of the river was dominated by transient bacteria and not polarized between transient and core bacteria as in the WWTP (S1 Fig). The number of ASVs detected in all the three tanks was 719 (46.0%), as shown in Fig 5C. Relative abundances of these common ASVs reached 0.834±0.121 SD in the anaerobic tank, 0.842±0.133 SD in the anoxic tank, and 0.867±0.124 SD in the aerobic tank. Consequently, the majority of core bacteria frequently detected over time were consistently detected in all three tanks, indicating a continuity in the temporal and spatial core bacteria. The stability of this dominant core community has significant implications for environmental health risks. Recent studies suggest that stable core taxa in WWTPs can serve as persistent reservoirs for antibiotic resistance genes (ARGs), facilitating their propagation regardless of transient influent fluctuations [32–34]. The polarization observed in this study, where a stable core dominates despite seasonal changes, suggests that if these core taxa harbor ARGs, the WWTP could act as a continuous source of resistomes released into the receiving river, necessitating long-term monitoring beyond standard water quality indices.

## Impact of treated wastewater on riverine bacterial communities

Wastewater is discharged into the rivers after nutrient removal and chlorine disinfection. The impact of bacterial DNA, in treated wastewater, on a river was evaluated using 16S rRNA gene analysis of river water upstream and downstream of the treated wastewater discharge (Fig 6, S3 Dataset). We identified 206 ASVs derived from bacterial DNA introduced into the river after the discharge of treated wastewater (detected at sites S2 and S3, but not at site S1) (Fig 6B). The changes in relative abundance of bacterial DNA at different sites are shown in Fig 6C. Relative abundances of the ASVs were 0.264±0.037 SD at site S2, 0.114±0.041 SD at site S3, and 0.052±0.009 SD at site S4, indicating a significant difference in the relative abundance between sites S2 and S4 ($p<0.05$, Friedman test; S2 vs S3: $p<0.05$, Nemenyi post-hoc test). These results suggest that the abundance of wastewater-derived bacteria decreased significantly as the water flowed downstream. Since there are no tributary inflows between sites S3 and S4, this reduction cannot be attributed to

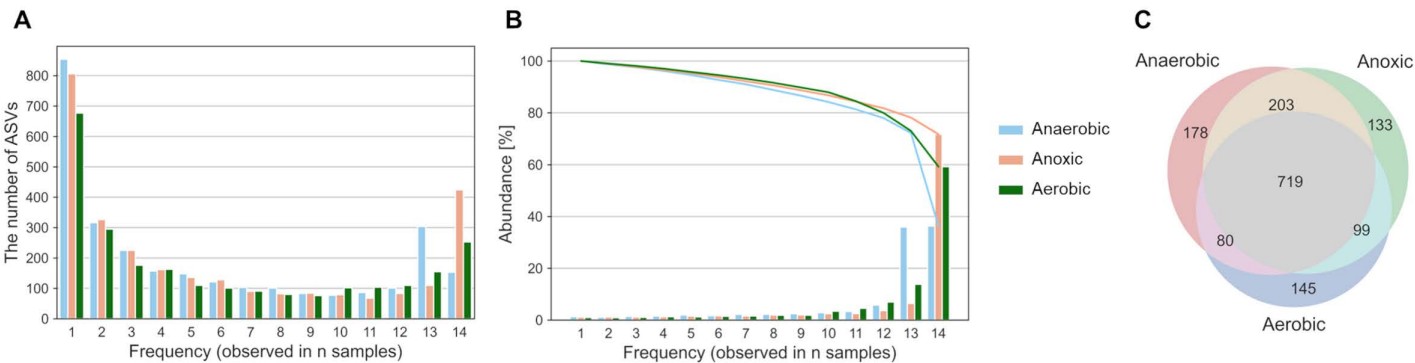

**Fig 5. Transient and core bacteria in A$_2$O treatment process. (A)** Relationship between the number of detections and the number of ASVs in 14 time-series samplings, and **(B)** their bacterial abundance. **(C)** Venn diagram of ASVs detected in anaerobic, anoxic, and aerobic tanks.

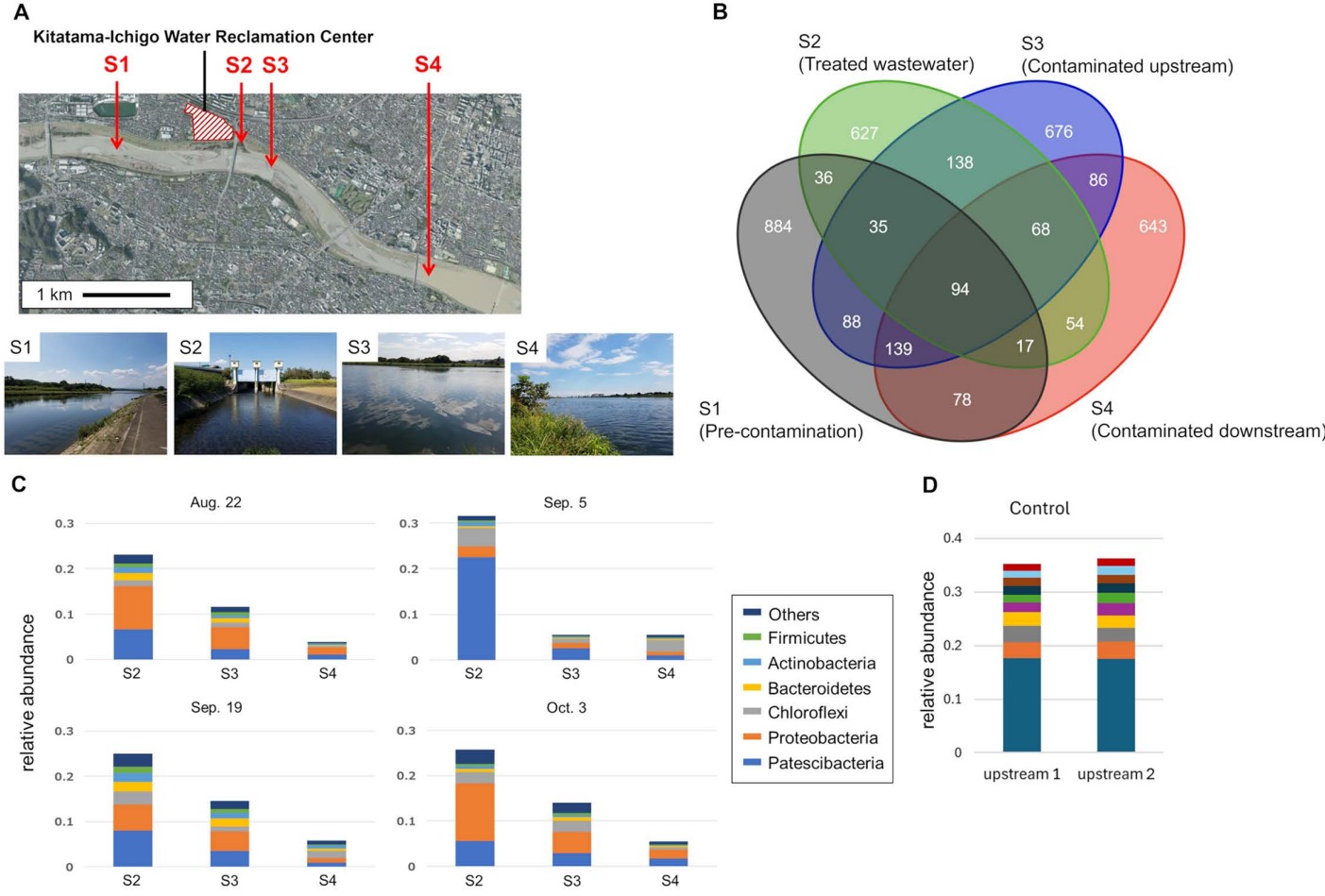

**Fig 6. Bacterial community characteristics before and after treated wastewater discharge. (A)** Site S1 is located before the treated wastewater discharge, site S2 is at the discharge point, and sites S3 and S4 are immediately after the discharge and 2 km downstream, respectively. The map was created by editing GSI Maps published by the Geospatial Information Authority of Japan [35] (https://maps.gsi.go.jp/#15/35.651333/139.507871/&base=ort&ls=ort&disp=1&vs=c1g1j0h0k0l0u0t0z0r0s0m0f1&d=m) under a CC BY license, original copyright 2025. **(B)** Venn diagram of ASVs detected at the four sites. **(C)** Total relative abundance of bacterial ASVs detected at sites S2 and S3 but not at site S1. The underlying data for these relative abundances are provided in S3 Dataset. **(D)** Comparison of the relative abundance of the top 10 dominant ASVs between two upstream reference sites (upstream 1 is located approximately 8.5 km upstream of upstream 2) using consistent color coding for each ASV to demonstrate natural background stability.

hydrological dilution from increased water volume. Instead, it is likely caused by physical dispersion (transitioning from incomplete mixing at S3 to complete mixing at S4) and removal processes. Notably, since the wastewater is treated with sodium hypochlorite before discharge, many detected sequences likely represent DNA from inactivated bacteria (relic DNA). Thus, the downstream decline reflects not only sedimentation but also the degradation of this DNA in the natural river environment.

To confirm whether the observed changes were due to the WWTP discharge or natural spatial variation, we compared bacterial communities between two upstream reference sites separated by 8.5 km (Fig 6D). The bacterial composition in the natural upstream river environment was highly stable; the total relative abundance of the top 10 dominant ASVs showed negligible change, accounting for 35.2% at upstream 1 and 36.2% at upstream 2. In contrast, the bacterial composition changed drastically at the WWTP discharge point (Site S3) compared to the upstream site (Site S1). This stability

in the upstream reference sites confirms that the community shift observed at Site S3 is a direct consequence of the treated wastewater discharge rather than natural fluctuations.

In this study, the abundance of wastewater-derived bacteria significantly decreased 2 km downstream of the treated-water discharge point, indicating that the influence of the effluent was spatially limited. This phenomenon was also observed in other rivers. Both the Riera de Cànove in Catalonia, Spain, and the Jialu in Zhengzhou City, China, exhibited a significant reduction in wastewater-associated bacteria 1 km downstream from the point of treated wastewater discharge [36,37]. In contrast, untreated wastewater discharged into a river does not lead to such mitigation, even 1 km downstream [38]. This implies that the rapid attenuation of wastewater impact observed in our study can be attributed to proper treatment of wastewater using the A$_2$O process. The WWTPs in Tokyo disinfect wastewater with chlorine before discharge, except during heavy rain overflows. In addition to disinfection, it may be necessary to remove microbial DNA from wastewater. Wastewater contains a variety of antibiotic resistance genes, raising concerns that horizontal gene transfer could lead to the emergence of antibiotic-resistant bacteria [39–42]. In the present study, we observed a decrease in microbial DNA in the treated water as it flowed downstream. However, previous studies have reported that the impact of treated water persists more in river sediments than in river water [36]. Therefore, it is necessary to investigate the influence on river sediment in future studies.

## Conclusions

This study investigated the bacterial dynamics of a wastewater treatment process and its receiving river to assess the effectiveness and ecological impact of wastewater treatment. The A$_2$O process effectively removed nutrients, although seasonal temperature changes influenced the nitrification. Decreased temperatures correlated with a decrease in *Nitrospira* abundance and potentially reduced nitrate production, highlighting the temperature sensitivity of the treatment process. The bacterial community in the wastewater treatment plant was polarized, with a significant portion of core bacteria consistently present over time and across different tanks, indicating a stable and functional bacterial ecosystem. The bacterial DNA introduced through the discharge of treated wastewater into the Tama River decreased as it flowed downstream, indicating dilution and dispersal. Understanding wastewater treatment processes using bacteria and their impact on riverine ecosystems is crucial for environmental conservation and preservation of water quality.

## Supporting information

**S1 Table. Tank capacity and residence time of each tank.**
(XLSX)

**S2 Table. Operational parameters of the A$_2$O process during the sampling period.** Representative monthly values for Oxidation-Reduction Potential (ORP) in anaerobic and anoxic tanks, Mixed Liquor Suspended Solids (MLSS) and Dissolved Oxygen (DO) in the aerobic tank, and influent temperature.
(XLSX)

**S3 Table. Geographic coordinates of the sampling sites.**
(XLSX)

**S1 Fig. Transient and core bacteria in Tama river.** (A) six samplings at eight sites (TR1–TR8). The map was created by editing GSI Maps published by the Geospatial Information Authority of Japan [35] (https://maps.gsi.go.jp/#11/35.629396/139.434357/&base=english&ls=english&disp=1&vs=c1g1j0h0k0l0u0t0z0r0s0m0f1&d=m) under a CC BY license, original copyright 2025. (B) Relationship between the number of detections and the number of ASVs converted to ratio in 6 time-series samplings. Series of eight bars from white to black represent sampling sites, with white indicating TR1 and black indicating TR8.
(TIF)

**S1 Dataset. Ion concentration data in the A$_2$O process.**
(XLSX)

**S2 Dataset. Bacterial relative abundance data in the A$_2$O process.**
(XLSX)

**S3 Dataset. Bacterial relative abundance data in the river.**
(XLSX)

## Acknowledgments

We thank the staff of the Tokyo Metropolitan Government Bureau of Sewerage for their cooperation in collecting wastewater samples and providing the operational data of the wastewater treatment plant.

## Author contributions

**Conceptualization:** Akifumi Nishida.

**Data curation:** Akifumi Nishida.

**Formal analysis:** Akifumi Nishida, Mayuko Nakagawa.

**Funding acquisition:** Akifumi Nishida, Masayuki Yamamura.

**Investigation:** Akifumi Nishida, Mayuko Nakagawa.

**Methodology:** Akifumi Nishida, Mayuko Nakagawa.

**Project administration:** Akifumi Nishida.

**Resources:** Akifumi Nishida.

**Software:** Akifumi Nishida.

**Supervision:** Akifumi Nishida, Masayuki Yamamura.

**Validation:** Akifumi Nishida.

**Visualization:** Akifumi Nishida.

**Writing – original draft:** Akifumi Nishida.

**Writing – review & editing:** Akifumi Nishida.

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
