## [Decision Letter · Decision Letter 0]

22 Oct 2025

We look forward to receiving your revised manuscript.

Kind regards,

Shijian Ge, Ph.D.

Academic Editor

PLOS ONE

Journal Requirements:

“AN was supported by Nippon Life Insurance Foundation (2018-17, http://www.nihonseimei-zaidan.or.jp/kankyo/02.html) and JSPS KAKENHI grants (22K17999 and 23KK0192, https://www.jsps.go.jp/english/e-grants/). MY was supported by Tokyu Foundation (2018-16, https://foundation.tokyu.co.jp/). The funders had no role in study design, data collection and analysis, decision to publish, or preparation of the manuscript.”

4. We note that Figure 6 in your submission contain copyrighted images. All PLOS content is published under the Creative Commons Attribution License (CC BY 4.0), which means that the manuscript, images, and Supporting Information files will be freely available online, and any third party is permitted to access, download, copy, distribute, and use these materials in any way, even commercially, with proper attribution. For more information, see our copyright guidelines: http://journals.plos.org/plosone/s/licenses-and-copyright.

a. You may seek permission from the original copyright holder of Figure 6 to publish the content specifically under the CC BY 4.0 license.

b . If you are unable to obtain permission from the original copyright holder to publish these figures under the CC BY 4.0 license or if the copyright holder’s requirements are incompatible with the CC BY 4.0 license, please either i) remove the figure or ii) supply a replacement figure that complies with the CC BY 4.0 license. Please check copyright information on all replacement figures and update the figure caption with source information. If applicable, please specify in the figure caption text when a figure is similar but not identical to the original image and is therefore for illustrative purposes only.

5. We note that Figure S2 in your submission contain map/satellite image which may be copyrighted. All PLOS content is published under the Creative Commons Attribution License (CC BY 4.0), which means that the manuscript, images, and Supporting Information files will be freely available online, and any third party is permitted to access, download, copy, distribute, and use these materials in any way, even commercially, with proper attribution. For these reasons, we cannot publish previously copyrighted maps or satellite images created using proprietary data, such as Google software (Google Maps, Street View, and Earth). For more information, see our copyright guidelines: http://journals.plos.org/plosone/s/licenses-and-copyright.

a. You may seek permission from the original copyright holder of Figure S2 to publish the content specifically under the CC BY 4.0 license.

Reviewers' comments:

Reviewer's Responses to Questions

**Comments to the Author**

1. Is the manuscript technically sound, and do the data support the conclusions?

Reviewer #1: Yes

Reviewer #2: Yes

2. Has the statistical analysis been performed appropriately and rigorously?

Reviewer #1: I Don't Know

Reviewer #2: I Don't Know

3. Have the authors made all data underlying the findings in their manuscript fully available?

Reviewer #1: Yes

Reviewer #2: No

4. Is the manuscript presented in an intelligible fashion and written in standard English?

Reviewer #1: Yes

Reviewer #2: Yes

Reviewer #1: This well-designed study provides valuable insights into bacterial dynamics in an A₂O wastewater treatment plant and its impact on the Tama River ecosystem. While the experimental approach and integration of microbial/chemical data are strengths, several areas require enhancement to clarify scientific foundations, methodological transparency, and environmental implications before publication.

See attached documents

Reviewer #2: Line 50, only NOB are mentioned, but to be correct also AOB should be mentioned too in the sentence, otherwise the statement is incomplete (or technically comammox nitrospira could be mentioned, but that is quite specific and not what the MS is about).

Line 52, I think it would be nice to mention the release of nitrous oxide here, but not strictly necessary if the authors prefer not to.

Line 68 to 70, the first half of the sentence simply states that these types of studies are important (without further elaboration), the second half then continues to state that it is also important for conservation and understanding of the bacterial communities along the river. Something seems to be missing or removed from the sentence to lead to this sentence structure. Please try to avoid stating that something is important without explaining why or for what.

Line 137, I think it would be better to report on the nitrogen removal efficiency (and potential S to) of the WWTP rather than just the figure showing the different concentrations. It gives the reader a clearer picture of the effectiveness of the WWTP.

Line 147, could the authors further elaborate on the interpretation of figure 3B? The temporal effects seem much stronger then the differences between the tanks, with for example 4s clustering close together, but for away from for example 13s which are on the other side of box.

Fig 6, How much natural variation is there when using ASVs to study the difference between bacterial communities in natural environments? Fig6 shows overlaps and unique ASVs for every sampling point, it would be interesting to see an additional sampling point that is not expected to be effected by the WWTP (say an S0) to see how much the overlap is with S1 as a consequence of natural variability rather then as a consequence of distance to the WWTP. This could be seen as a control sampling location. While I understand that it is now too late to redesign the study at this level, this question keeps me wondering and makes it more difficult to know how much of the observed differences are due to the distance from the WWTP and how much is due to natural variability. I think it would improve the manuscript if the authors could reflect on this.

Line 218, are they diluted or do other processes also play a role here? How much dilution do we expect on a few km of river, are there any tributary inflows? Is the effect caused by incomplete mixing of river and WWTP at S3 causing a higher fraction of the sample to be coming directly from the WWTP whereas S4 is better mixed? Could other processes explain the observation? Sedimentation/flocculation, environmental or competitive selection, predation?

Line 227, please define what you mean with a recovered bacterial community. The community is still clearly different from S1 (which can also be seen in Fig 6B) what conditions need to be met to consider the community recovered?

The data is available through the figures which is a nice way to present the data, however this also leaves the readers guessing for more exact values as it is difficult to know the exact intensity of a color gradient or the precise height of a bar in a bar chart. It would be better if this data is also available in the SI.

.

Reviewer #1: **Yes:** Dr. Oussama BOUOUAROURDr. Oussama BOUOUAROURDr. Oussama BOUOUAROURDr. Oussama BOUOUAROUR

Reviewer #2: No

---

## [Author Response · Author response to Decision Letter 1]

14 Jan 2026

January 1, 2026

PONE-D-25-23321

The role of bacteria in wastewater treatment and their impact on riverine bacterial ecosystems

PLOS ONE

Dear Dr. Shijian Ge

We appreciate you and the reviewers to review our article and give us a chance to improve the article more. The insightful comments have strengthened the article. It is great pleasure that we submit the article for further consideration. The manuscript has been carefully rechecked, and appropriate changes have been made in accordance with the suggestions of the editor and the reviewers.

Our detailed point-by-point responses to the comments are attached to this letter. In the revised manuscript, all changes have been tracked using the "Track Changes" function in Microsoft Word. All the authors have revised and approved the new version of the manuscript.

We hope that the revised manuscript is now deemed suitable for publication in your esteemed journal, PLOS ONE.

I look forward to your reply.

Sincerely,

Akifumi Nishida

Department of Molecular Microbiology,

Tokyo University of Agriculture,

Tokyo 156-8502, Japan.

+81-90-7385-5684

an207919@nodai.ac.jp

Point-by-point responses to the reviewer’s comments

Response to Journal Requirements

We have carefully reviewed the PLOS ONE style guidelines and formatted our manuscript, including the title page, affiliations, and file naming, according to the templates provided.

In accordance with your request, we have amended our Funding Statement to include the required sentence: “There was no additional external funding received for this study.”. We have also included this full amended statement in our cover letter.

3. Please include your full ethics statement in the ‘Methods’ section of your manuscript file.

We have included a full ethics statement at the beginning of the ‘Methods’ section in the revised manuscript (Lines 86–94). We have explicitly stated the approvals obtained from the Bureau of Sewerage, Tokyo Metropolitan Government. We also clarified that informed consent was not required as the study did not involve human participants.

4 and 5. We note that Figure 6 and Figure S2 in your submission contain copyrighted images. We require you to either (1) present written permission or (2) remove the figures.

We have replaced the figures containing maps (Fig 6 and the original Fig S2) with those created using GSI Maps published by the Geospatial Information Authority of Japan. These maps are provided under a license compatible with CC BY 4.0 (https://www.gsi.go.jp/ENGLISH/index.html). Please note that due to the relocation of the previous Figure S1 into the main text, the original Figure S2 has been renumbered as Figure S1 in the revised Supporting Information. The figure captions have been updated to: “The map was created by editing GSI Maps published by the Geospatial Information Authority of Japan under a CC BY license, original copyright 2025.”

Response to Reviewer #1

Major Revisions

1. The Introduction should explicitly state the central hypothesis rather than implying objectives (e.g., lines 70-74), with a concise formulation such as: "We hypothesized that seasonal temperature variations would reduce nitrifier activity in the A₂O process,……" to frame the study's scientific intent and connect it to broader ecological risks like antibiotic resistance gene dispersal.

We appreciate this valuable suggestion. In the revised Introduction, we have reformulated the final paragraph to explicitly state our hypotheses regarding temperature-driven shifts in nitrifier abundance and the spatial attenuation of wastewater-derived bacteria (Lines 75–83).

2. Statistical methodologies need expanded justification, particularly the selection of the use of Friedman/Nemenyi tests for riverine data without explaining non-parametric choices); a dedicated "Statistical Analysis" subsection should clarify software parameters, permutation counts for PERMANOVA, and p-value thresholds while justifying similarity metrics.

We have added a dedicated "Statistical Analysis" subsection (Lines 144–163). We justified the use of non-parametric tests (Friedman and Nemenyi) based on the non-normal distribution of our data (Shapiro-Wilk test). We also specified the similarity metrics (Weighted UniFrac) and parameters for PERMANOVA (999 permutations) with the threshold (p < 0.05).

3. Environmental context must be strengthened by integrating ancillary variables (e.g., pH, dissolved oxygen, organic load) into multivariate models like PCoA 3B) to avoid overreliance on temperature when explaining nitrification shifts and river recovery (lines while discussing alternative drivers such as hydraulic retention time versus temperature for nitrification efficiency and sediment interactions for DNA persistence.

We have included operational data for ORP, MLSS, and DO in the revised manuscript (Table S2). Analysis of these data ruled out oxygen deficiency as a driver, while MLSS trends supported our hypothesis that temperature-dependent kinetics were the primary driver of nitrification shifts (Lines 229–238).

4. Contradictory findings require deeper interpretation, specifically the decline in Nitrospira abundance alongside increased NO₃⁻ production and implications of core-transient polarization; the Discussion should address microbial functional redundancy (e.g., Sulfuritalea's compensatory role) and connect core bacterial stability to antibiotic resistance risks using recent literature.

We have revised the "Results and discussion" section to explain this based on substrate-limited kinetics. As ammonium removal efficiency remained high (98.4 ± 5.8%), the nitrifying community retained sufficient capacity; thus, increased NO₃⁻ was a stoichiometric result of higher winter influent ammonium loads (Lines 223–228).

Minor Revisions

1. Figures need enhanced clarity: Label functional guilds on y-axes in Fig. 4 and define color keys for temporal trends, add geographic coordinates and scale bars to the river map in Fig. 6A, and incorporate α-diversity statistics from S1 Fig into the main text to support tank comparisons.

We have updated Figure 4 with high-contrast colors and moved α-diversity plots from Figure S1 to the main text (Fig 3B). We have added the geographic coordinates as Table S3.

2. Language requires standardization and polishing: Professional English editing is recommended to address grammatical errors.

The manuscript has been professionally edited by Editage for English language and clarity.

3. References need reorganization and formatting: Sort chronologically to highlight thematic progression (e.g., group nitrification studies [22-24,26]), italicize genus/species names (e.g., Nitrospira), and ensure DOI consistency (reference 18 currently lacks DOI).

References have been reorganized chronologically, and genus names italicized.

4. Data accessibility should be clarified: Explicitly state whether sediment data referenced in lines 239-240 are archived in repositories.

We have clarified that the sediment data mentioned in the text refers to findings from previous literature (Lines 333–335).

Response to Reviewer #2

1. Line 52, I think it would be nice to mention the release of nitrous oxide here, but not strictly necessary if the authors prefer not to.

We have now included information regarding both ammonia-oxidizing bacteria (AOB) and nitrite-oxidizing bacteria (NOB) to provide a complete description of the nitrification process (Lines 51–53).

2. Line 52, I think it would be nice to mention the release of nitrous oxide here.

We have added a mention of nitrous oxide generation under specific conditions during the nitrogen removal process (Lines 54–57).

3. Line 68 to 70, the first half of the sentence simply states that these types of studies are important (without further elaboration), the second half then continues to state that it is also important for conservation and understanding of the bacterial communities along the river. Something seems to be missing or removed from the sentence to lead to this sentence structure. Please try to avoid stating that something is important without explaining why or for what.

We have revised the sentence to clearly connect previous studies with the motivation of our work—specifically, the evaluation of potential ecological risks such as the dispersal of antibiotic-resistant bacteria (Lines 71–74).

4. Line 137, I think it would be better to report on the nitrogen removal efficiency.

We now report the NH₄⁺ removal (nitrification) efficiency in the aerobic tank, which was 98.4 ± 5.8% (mean ± SD). We added a short explanation to clarify that this metric appropriately reflects the functional performance of the nitrification step (Line 178).

5. Line 147, could the authors further elaborate on the interpretation of figure 3B? The temporal effects seem much stronger.

We acknowledge the strong temporal stratification driven by temperature, as confirmed by our vector analysis (p < 0.05). We have revised the text to explicitly discuss this seasonal dominance while noting that PERMANOVA still confirms significant differences between tanks (Lines 188–195).

6. Fig 6, How much natural variation is there when using ASVs to study the difference between bacterial communities in natural environments? Fig6 shows overlaps and unique ASVs for every sampling point, it would be interesting to see an additional sampling point that is not expected to be effected by the WWTP (say an S0) to see how much the overlap is with S1 as a consequence of natural variability rather then as a consequence of distance to the WWTP. This could be seen as a control sampling location. While I understand that it is now too late to redesign the study at this level, this question keeps me wondering and makes it more difficult to know how much of the observed differences are due to the distance from the WWTP and how much is due to natural variability. I think it would improve the manuscript if the authors could reflect on this.

We incorporated additional data from two upstream control sites (separated by 8.5 km) as Fig 6D. The stability of the bacterial community at these sites confirms that the shifts observed at Site S3 are a direct consequence of WWTP discharge, not natural fluctuations (Lines 299–307).

7. Line 218, Line 218, are they diluted or do other processes also play a role here? How much dilution do we expect on a few km of river, are there any tributary inflows? Is the effect caused by incomplete mixing of river and WWTP at S3 causing a higher fraction of the sample to be coming directly from the WWTP whereas S4 is better mixed? Could other processes explain the observation? Sedimentation/flocculation, environmental or competitive selection, predation?

We have clarified that the reduction in wastewater-associated ASVs results from physical mixing dynamics, sedimentation, and the degradation of relic DNA from disinfected bacteria, rather than simple hydrological dilution (Line 291–298).

8. Line 227, please define what you mean with a recovered bacterial community. The community is still clearly different from S1 (which can also be seen in Fig 6B) what conditions need to be met to consider the community recovered?

We have removed the term "recovered" and now use "attenuation of the wastewater impact" to more accurately describe the significant decrease in wastewater-derived ASVs at the downstream site (Line 326).

9. The data is available through the figures which is a nice way to present the data, however this also leaves the readers guessing for more exact values as it is difficult to know the exact intensity of a color gradient or the precise height of a bar in a bar chart. It would be better if this data is also available in the SI.

We have provided additional supporting datasets (Dataset S1–S3) containing the exact numerical values to ensure full data accessibility.

Dataset S1 contains the exact ion concentration values measured in the A2O process, which correspond to the data visualized in Figure 2.

Dataset S2 provides the bacterial relative abundance data in the A2O process at the phylum and genus levels, corresponding to Figures 3 and 4.

Dataset S3 includes the relative abundance data of wastewater-derived ASVs in the Tama River, which serves as the basis for Figure 6C.

---

## [Decision Letter · Decision Letter 1]

18 Mar 2026

The role of bacteria in wastewater treatment and the impact of treated wastewater on riverine bacterial ecosystems

PONE-D-25-23321R1

Dear Dr. Nishida,

We’re pleased to inform you that your manuscript has been judged scientifically suitable for publication and will be formally accepted for publication once it meets all outstanding technical requirements.

Kind regards,

PLOS One

Reviewers' comments:

Reviewer's Responses to Questions

**Comments to the Author**

Reviewer #1: All comments have been addressed

2. Is the manuscript technically sound, and do the data support the conclusions?

Reviewer #1: Yes

3. Has the statistical analysis been performed appropriately and rigorously?

Reviewer #1: Yes

4. Have the authors made all data underlying the findings in their manuscript fully available?

Reviewer #1: Yes

5. Is the manuscript presented in an intelligible fashion and written in standard English?

Reviewer #1: Yes

Reviewer #1: This manuscript presents a well-designed study investigating bacterial dynamics within an A₂O wastewater treatment plant and their downstream effects on the River ecosystem. The authors have been highly responsive to the previous round of review comments, and the revised manuscript demonstrates substantial improvements in clarity, methodological transparency, and scientific rigor. I appreciate the thoroughness with which they have addressed each concern raised.

.

Reviewer #1: **Yes:** Dr Oussama BOUOUAROURDr Oussama BOUOUAROURDr Oussama BOUOUAROURDr Oussama BOUOUAROUR

---

## [Editor Report · Acceptance letter]

PONE-D-25-23321R1

PLOS One

Dear Dr. Nishida,

I'm pleased to inform you that your manuscript has been deemed suitable for publication in PLOS One. Congratulations! Your manuscript is now being handed over to our production team.

Kind regards,

on behalf of

Professor Shijian Ge

Academic Editor

PLOS One